# Genome-Wide Identification and Analysis of U-Box E3 Ubiquitin-Protein Ligase Gene Family in Banana

**DOI:** 10.3390/ijms19123874

**Published:** 2018-12-04

**Authors:** Huigang Hu, Chen Dong, Dequan Sun, Yulin Hu, Jianghui Xie

**Affiliations:** Key Laboratory of Tropical Fruit Biology, Ministry of Agriculture, South Subtropical Crop Research Institute, Chinese Academy of Tropical Agricultural Science, Zhanjiang 524091, China; zjhuhuigang@163.com/huhuigang@sina.com (H.H.); nysdongchen1981@163.com (C.D.); sscrisun@163.com (D.S.); huyulin2003@126.com (Y.H.)

**Keywords:** banana, ubiquitin-protein ligases, gene family, fruit development, abiotic stresses

## Abstract

The U-box gene family is a family of genes which encode U-box domain-containing proteins. However, little is known about U-box genes in banana (*Musa acuminata*). In this study, 91 U-box genes were identified in banana based on its genome sequence. The banana U-box genes were distributed across all 12 chromosomes at different densities. Phylogenetic analysis of U-box genes from banana, *Arabidopsis*, and rice suggested that they can be clustered into seven subgroups (I–VII), and most U-box genes had a closer relationship between banana and rice relative to *Arabidopsis*. Typical U-box domains were found in all identified MaU-box genes through the analysis of conserved motifs. Four conserved domains were found in major banana U-box proteins. The MaU-box gene family had the highest expression in the roots at the initial fruit developmental stage. The MaU-box genes exhibited stronger response to drought than to salt and low temperatures. To the best of our knowledge, this report is the first to perform genome-wide identification and analysis of the U-box gene family in banana, and the results should provide valuable information for better understanding of the function of U-box in banana.

## 1. Introduction

The ubiquitin/26S proteasome (UPS) pathway degrades ubiquitinated substrate proteins and is extensively involved in various cellular processes [1]. The diverse aspects of plant growth and development and the degradation of short-lived regulatory proteins can be regulated by the UPS [2,3,4]. E1 Ub-activating enzyme, E2 Ub-conjugating enzyme, and E3 Ub ligase are necessary for ubiquitin activation and transfer [5]. First, E1 activates the ubiquitin molecule in an ATP-dependent manner, and then E2 facilitates the attachment of ubiquitin molecule to the target protein in the presence of E3 [6]. E3 ligase plays an important role in protein ubiquitination because E3 can identify target proteins for modification [7]. A single protein or a protein complex joins the ubiquitin reaction, which could be conferred by E3 ligase [8,9]. Ubiquitin E3 ligases facilitate the covalent attachment of ubiquitin to target proteins in eukaryotes [10]. HECT, RING finger, and U-box domain proteins are three types of single-protein E3 ligases [11]. U-box proteins are found in yeast, plants, and animals [12,13,14,15]. The U-box domain is composed of approximately 75 amino acids (aa) [16,17]. Many U-box proteins had function of E3 ligases [18,19]. The genome of *Arabidopsis thaliana* has more than 60 U-box genes, which have many functions in plants [16]. A previous study has identified the functions of U-box E3 ligases in parsley, tomato, tobacco, and rice [20]. OsU-box gene 51 negatively regulates cell death signaling according to cell death assay [17]. The U-box E3 ligase NtACRE276 of tobacco may play a role in Cf9/Avr9-elicited defense [21]. On the basis of protein domains, eight groups of Plant U-box (PUB) genes are present in the 125 PUB genes of soybean [22]. The flowering condition could be changed in GmPUB8-overexpressing *Arabidopsis*, which flowered earlier under middle- and short-day conditions but later under long-day conditions [22]. Inactivation of the Arabidopsis PUB13 also results in spontaneous cell death, enhanced levels of the defence hormone SA, and early flowering [23]. In grapevine, the PUB gene significantly regulates the accumulation of resistance proteins under both biotic and abiotic stresses [20].

Banana (*Musa* spp.) is one of the world’s most important fruits [24,25]. The sequencing of the whole genome of banana (*Musa acuminata*) provides a good platform for the development of banana molecular biology [26]. Until now, the U-box gene family of banana is rarely studied. U-box genes may play important roles in the growth and development of banana, so investigating the E3 gene family in banana is necessary. In this study, the whole genome of the banana U-box gene was determined and analyzed. The conserved domain structure, subgroup classification, evolutionary relationship, intron and exon structure, gene expansion, chromosome mapping, and expression profile analysis were studied, providing a theoretical basis for the analysis of U-box gene functions.

## 2. Results

### 2.1. Identification and Chromosomal Localization of U-Box Gene Family Members

In this study, 91 PUB genes are found in banana genome (Table 1). The MaU-box protein contains a 60–70 aa U-box conserved domain. The length of MaU-box was from 660 (MaU-box69) to 6279 bp (MaU-box57), and the average length was 1789 bp. The predicted protein product range was 219–2092 aa, with an average length of 595 aa. The relative molecular weight (MW) ranged from 23.38 kD to 223.93 kD, with an average of 64.85. The isoelectric point (PI) was in the range of 4.96 (MaU-box78) to 9.57 (MaU-box13). Subcellular localization analysis indicated that 93% of the MaU-box proteins were located in the nucleus and that only six were located in the cytoplasm (Table 1). These findings suggested that the vast majority of MaU-box function in the nucleus.

A MaU-box chromosomal localization map was plotted (Figure 1). Ninety genes from 91 MaU-box genes were located on chromosomes. Chromosome 3, where the largest number of MaU-box genes was found, contained 11 MaU-box genes. It is followed by chromosomes 4, 5, and 11, which contained 10 MaU-box genes. Nine MaU-box genes were located in chromosomes 7, 9, and 10; 7 MaU-box genes were found in chromosome 8; 6 MaU-box genes were observed in chromosome 1; 5 MaU-box genes were localized in chromosome 2; and only 4 MaU-box genes were detected in chromosome 11.

### 2.2. Gene Structure and Phylogenetic Analysis of U-Box Gene Family Members

By comparing the full-length cDNA sequence with the corresponding genomic DNA sequence, the exon–intron structure of each MaU-box was determined. The number of exons in MaU-box genes ranged from 1 to 18 (Figure 2).

To study the evolutionary relationship of banana U-box proteins, a neighbor-joining (NJ) tree was constructed with U-box proteins from banana, rice, and *Arabidopsis* (Figure 3). The aa sequences of the U-box of 91 proteins from banana, 61 from *Arabidopsis*, and 77 from rice were used. Phylogenetic analysis showed that all identified U-box proteins from banana together with *Arabidopsis* and rice were clearly divided into seven subgroups. Subgroups I, II, III, IV, V, VI, and VII contain 8, 2, 10, 8, 26, 32, and 5 gene family members, respectively. In general, the U-box from banana had a closer relationship with rice compared with *Arabidopsis*. Interestingly, these MaU-box genes with similar genetic structures are clustered together. For example, MaU-box51/65/84/91 of subgroup I each contain 11 exons, MaU-box66/87 of subgroup II each contain 18 exons, and MaU-box4/19/20/39/45/54 of subgroup III each contain 1 exon.

### 2.3. Analysis of MaU-Box Gene Family Conserved Motifs

To investigate the structural diversity and predict the function of MaU-box proteins, 20 conserved motifs in banana U-box were identified using the MEME motif search tool and annotated using SMART tools (Figure 4 and Figure 5). Among the 91 U-box genes, 45 (50%) contained U-box conservative motifs without ARM motifs, 22 (24%) contained ARM conservative motifs without U-box motifs, while 24 (26%) contained both U-box conserved motifs and ARM conserved motifs (Figure 6). Motifs 1 and 2 are U-box conservative motifs; motifs 5, 8, 10, 13, 16, and 18 are ARM conservative motifs; motifs 9 and 14 are Pkinase-Tyr motifs; and motif 6 is an STYKc motif. Motifs 3, 4, 7, 11, 12, 15, 17, 19, and 20 are unknown. The features of 20 motifs are shown in Figure 5.

### 2.4. Expression Profile of MaU-Box Genes in Different Organs

Figure 7 shows that among the 91 MaU-box genes, 88 were at least expressed in one tissue, occupying 97% of all gene numbers. Moreover, among the 88 genes, 76 were expressed in all tissues. The Mau-box gene family was differentially expressed in various tissues. This gene family had the highest expression in the roots, where 48 genes exhibited the highest expression. By contrast, the lowest expression was observed in the male flowers, where 17 genes (MaU-box8/9/13/16/17/19/21/28/30/32/35/37/38/46/50/56/89) exhibited the lowest expression.

### 2.5. Expression Profile of MaU-Box Genes in Fruit Developmental Period

Figure 8 illustrates that 62 MaU-box genes were at least expressed during one developmental phase, 59 of which were expressed in all developmental phases. The MaU-box gene family was also differentially expressed at various developmental phases of banana. The gene family had the highest expression at the beginning of banana’s development phase (day 25), during which 29 genes (MaU-box3/4/8/9/10/13/17/18/19/20/22/25/27/28/34/38/39/41/43/48/49/50/51/52/56/59/61/63/70) had the highest expression. High MaU-box gene expression was also observed during banana fructescence (day 88), which was second only to the expression on day 25. During the maturity phase of fructescence, 21 genes (MaU-box5/7/12/13/16/21/33/42/47/55/57/65/66/67/69/71/81/84/86/87/88) had the highest expression. Four genes had significant linear expression at the developmental phase. Among them, three genes (MaU-box8/27/61) were gradually downregulated, whereas one gene (MaU-box71) was gradually upregulated with increasing development time.

### 2.6. Differential Expression of MaU-Box Genes under Abiotic Stresses

Figure 9 shows that 60 MaU-box genes responded to drought, salt, and low-temperature stressors. Among these stressors, the MaU-box gene family showed the strongest response to drought. A total of 55 MaU-box genes exhibited the highest expression under this stressor, during which 45 genes were upregulated by more than tenfold. The MaU-box gene family showed the highest expression at 24 h, during which 54 genes exhibited the highest expression. Salt stress also resulted in the high regulation of the MaU-box gene family, and this stressor led to the highest expression of four genes (MaU-box63/65/71/78) and the upregulation of two genes (MaU-box63/65) by more than tenfold (*p* < 0.05).

## 3. Discussion

The characteristics and functions of the U-box gene family has been studied in several plants [27,28]. In the present study, systematic phylogenetic analyses were conducted to obtain a detailed classification and nomenclature of the banana U-box. We found 91 PUB (Plant U-box) genes in banana genome. Similarly, 61 U-box proteins of *Arabidopsis* [12] and 77 U-box-containing proteins of rice had been identified and analyzed [17]. In total, 125 soybean PUB (GmPUB) genes, which encode proteins containing the U-box domain, have been identified [22]. The distribution of U-box proteins among species of different kingdoms is uneven [17]. Our data showed that the banana U-box genes were distributed across all 11 chromosomes at different densities. Phylogenetic analysis of the U-box from banana, *Arabidopsis*, and rice suggested that the U-box could be clustered into seven subgroups (I–VII). A similar study in soybean found that 125 GmPUB proteins were classified into six groups by phylogenetic analysis [22]. In this study, most banana U-box proteins show closer phylogenetic distance to their putative banana homologs than to their corresponding putative rice and Arabidopsis orthologs. Moreover, the U-box from banana had a closer relationship with rice compared with Arabidopsis. Interestingly, banana proteins MaU-box56, MaU-box78, MaU-box83 and MaU-box84 showed a closer phylogenetic relationship to the rice proteins than to their banana paralogs, suggesting that these banana proteins and their corresponding rice orthologs have evolved from a common ancestor before the speciation of the two species [17]. In the present study, conserved motif analysis showed that all identified MaU-box had typical U-box domains. Generally, a protein–protein interaction domain in E3 ubiquitin ligases interacts with their substrates for ubiquitination [29], and a complete U-box domain was found in all PUB proteins [30,31,32]. The proteins that contained conserved motifs had low sequence similarity, suggesting that mutations were accumulated during evolution [22]. The U-box in banana are found in combination with a variation of domains including armadillo (ARM) repeats, WD40 repeats, the tetratricopeptide (TPR) domain. The ARM repeats have been shown mostly to mediate the interaction with substrates, indicating that interaction renders substrates available for ubiquitination [23]. So the U-box proteins without ARM repeats in banana might have different interactions of E3 ubiquitin ligases with their substrates compared with the U-box proteins containing ARM repeats. The MaU-box gene family was differentially expressed in various tissues of banana. Similarly, several AtPUB-ARM genes were widely expressed in different tissues [33]. The MaU-box gene family had the highest expression in the roots. In a previous study, 12 MaE2 genes had the highest expression levels in roots [24]. These results suggested that MaU-box genes might be involved in the formation of the root system. PUB proteins play important roles in regulating plant growth and development [34]. In the present study, the 29 MaU-box genes (MaU-box3/4/8/9/10/13/17/18/19/20/22/25/27/28/34/38/39/41/43/48/49/50/51/52/56/59/61/63/70) had the highest expression at the beginning of banana’s developmental stage (day 25), which could be explained by the fact that the highly expressed genes usually play important roles in plant development [35], suggesting ubiquitination activation through the first stages of fruit development. Of note, the expression of eight MaE2 genes [23] and three MaU-box genes decreased gradually with prolonged developmental time. In strawberry fruit, all the genes decreased gradually after the flowering stage [36]. These data indicated that some genes (e.g., the MaU-box gene) might play important roles for the growth and development of fruits. Studies have shown that the U-box protein is involved in the response to various environmental stresses [9,20,37]. The U-box protein gene quickly responded to both biotic stress and abiotic stress and significantly influenced the accumulation of resistance related proteins in grapevine [20]. Silencing tomato U-box E3 ligase ACRE74 lead to break down of Cf9-especified resistance against Cladosporium fulvum leaf mold [20]. The U-box genes of rice might be involved in the defense against diseases [17]. A previous study observed differential expression patterns in nine soybean genes under drought stress [22]. In the present study, the MaU-box genes exhibited stronger response to drought than to salt and low temperature. Under drought stress, 45 MaU-box genes were upregulated by more than tenfold. OsPUB57 showed a stronger expression only in the resistant plants carrying the Pi9-resistant gene [17]. Consistent with this study, in our study, the MaPUB84 and MaPUB91 genes which showed closer phylogenetic distance to OsPUB57 had high expressions under stress. These results indicated that PUB genes might have key functions in responding to drought stress in plant.

## 4. Materials and Methods

### 4.1. Plant Materials and Treatment

The test material “Brazil” banana was obtained from the banana plantation of the National Banana Industry Technical System of Zhanjiang Comprehensive Test Station, South Subtropical Crops Institute, Chinese Academy of Tropical Agricultural Sciences (Zhanjiang, Guangdong, China). Different organs (roots, stems, leaves, female flowers, male flowers) were collected to study the temporal and spatial expression patterns of bananas. The fruits were collected at different developmental stages (25, 45, 65, and 85 days after florescence) to study fruit development. A healthy and consistent banana seedling with four leaves was selected for stress experiments. The banana seedlings were treated with 20% PEG 6000 (drought stress treatment) and 200 mM NaCl (salt stress treatment) and harvested at different time points (1, 6, and 24 h) after treatment [23]. The experiments were performed in triplicate. All samples were frozen in liquid nitrogen and stored at −80 °C for the purpose of extracting RNA for expression analysis.

### 4.2. Genome Identification of Banana U-Box Gene Family Members

To identify the potential members of the banana U-box protein family, publish the Arabidopsis thaliana and rice U-box protein sequences as seed sequences, and used BLASTP method search the banana genome database (Banana Genome Hub, available online: http://banana-genome.cirad.fr/content/download-dh-pahang) and Phytozome (available online: http://www.phytozome.net/) database. All candidate U-box genes were further verified by using SMART conserved domain search tools (available online: http://smart.embl-heidelberg.de/), eliminating repeat sequences, and deleting genes without the U-box domain. The MW and PI prediction of all U-box proteins was performed using the ProtParam tool (available online: http: //web.expasy.org/orgparam/). Information on the MaU-box gene, including chromosomal location, DNA sequence, CDS sequence, and aa length, was obtained from phyome12 (available online: https://phytozome.jgi.doe.gov/pz/portal.html#!info?alias=Org_Gmax). The MW and theoretical PI of the candidate MaU-box protein were obtained using the ExPASy Online Tool (available online: http://expasy.org/tools/). The subcellular localization of banana U-box protein was predicted by using the online software, Plant-mPLoc (available online: http://www.csbio.sjtu.edu.cn/bioinf/plant-multi/#). Finally, chromosome mapping was performed using the MapInspect tool according to the position of the U-box on the chromosome. For convenience, the MaU-box genes were numbered MaU-box1–MaU-box91 according to the order of chromosome 1 to 11. The structure of each gene was visualized using the Gene Structure Display Server (available online: http://gsds.cbi.pku.edu.cn/).

### 4.3. MaU-Box Protein Conserved Motif and Phylogenetic Analysis

The protein conserved motif of the MaU-box gene family was analyzed using MEME Suite 4.11.4 (available online: http://meme.nbcr.net/meme/) software. The maximum number of protein motifs was 20, and the length of the motifs was 6 to 200 aa.

To understand the evolutionary relationship of the U-box gene, we used the Clustal X version 1.83 software (lllkirch, France) with default parameters to compare the sequences of *Arabidopsis thaliana*, rice, and banana U-box gene family members. The phylogenetic tree was constructed by comparing the results with MEGA6.0 software (state college, PA, USA). The parameters of the software were set as follows: NJ method as the adjacency method and Poisson correction, paired delete, and bootstrap (1000 repetitions).

### 4.4. Gene Expression Analysis

The MaActin fragment of the banana was selected as the internal reference, and the primers were designed according to the registered sequence. All the MaU-box genes secific primers were designed according to the coding sequences by Primer5 software (PREMIER Biosoft International, Palo Alto, CA, USA) and checked using Blast in NCBI (available online: https://www.ncbi.nlm.nih.gov/). The relative expression level of the U-box gene was calculated using Equation 2^−ΔΔ^*C*t.

## 5. Conclusions

Ninety-one U-box genes of the banana genome were classified into seven subgroups. Typical U-box domains were found in all identified MaU-box. The MaU-box gene family had the highest expression in the roots, and the strongest expression was found at the first developmental stage. The MaU-box genes exhibited stronger response to drought than to salt and low temperature. The results of this study provide information on the evolution and functions of the MaU-box genes.

## Figures and Tables

**Figure 1 ijms-19-03874-f001:**
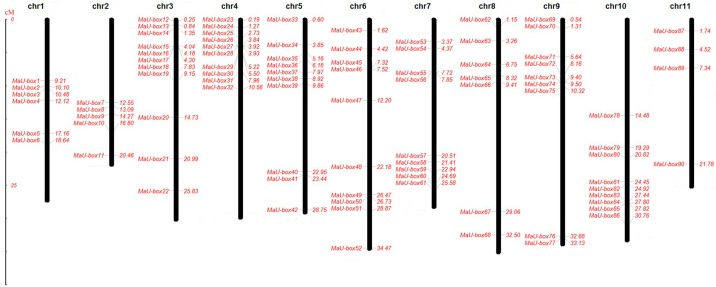
MaU-box chromosomal localization maps. The size of a chromosome is indicated by its relative length in centimorgan (cM).

**Figure 2 ijms-19-03874-f002:**
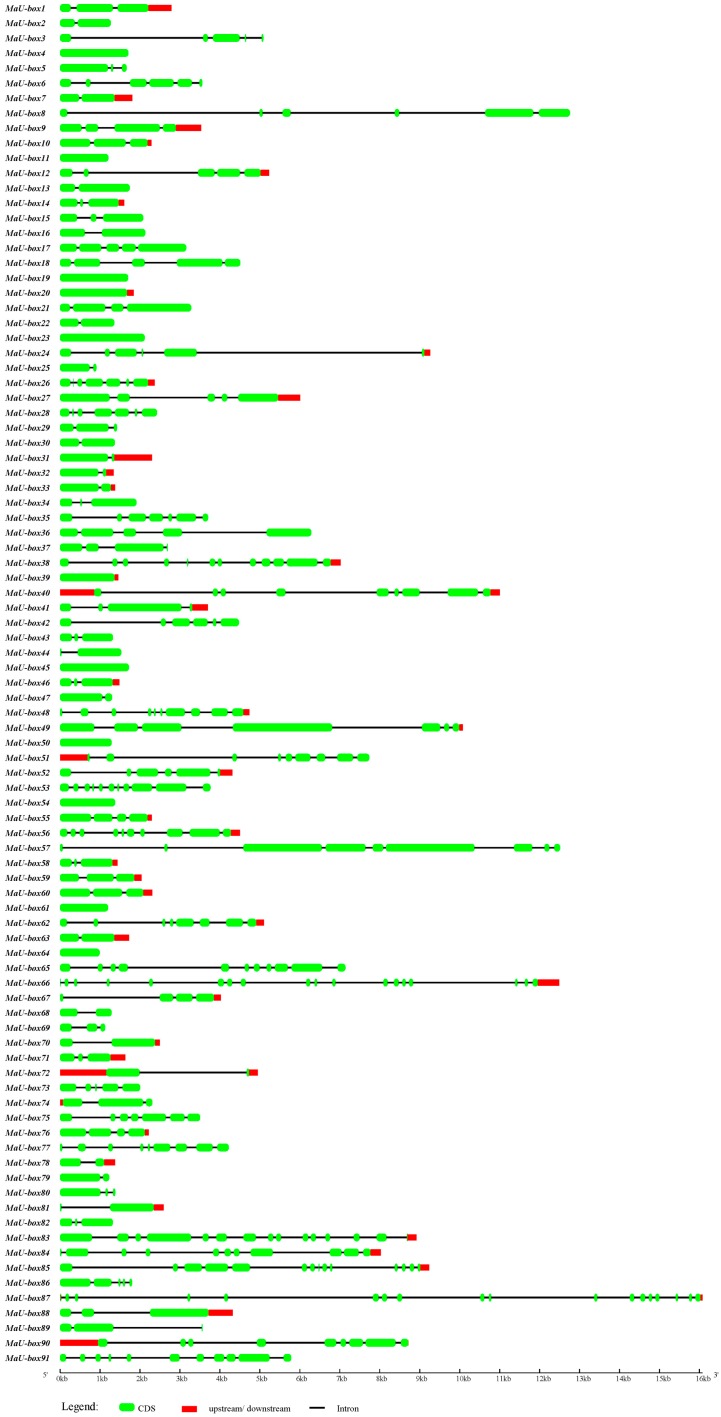
MaU-box gene structures.

**Figure 3 ijms-19-03874-f003:**
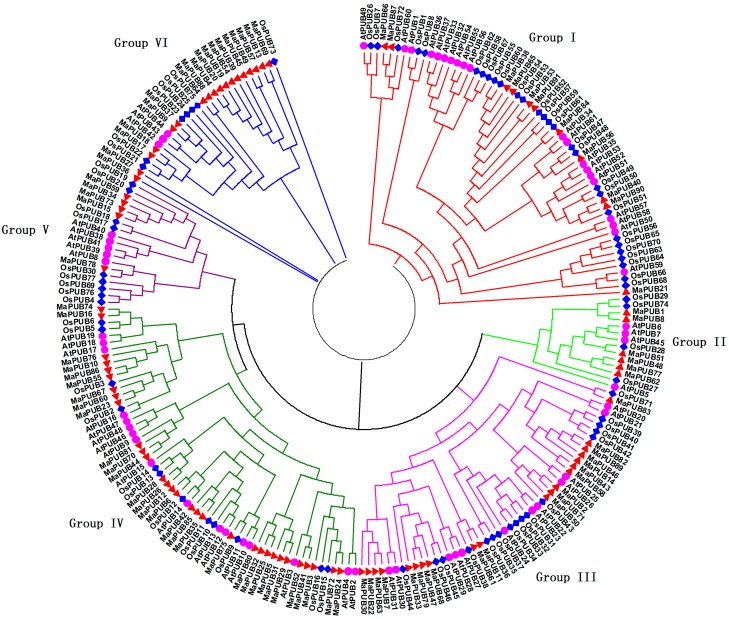
Phylogenetic tree representing relationships among U-box domains of banana, Arabidopsis and rice. The different colors and different numbers (Group I–VI) indicate different MaPUB subgroups. The different shapes indicate different species.

**Figure 4 ijms-19-03874-f004:**
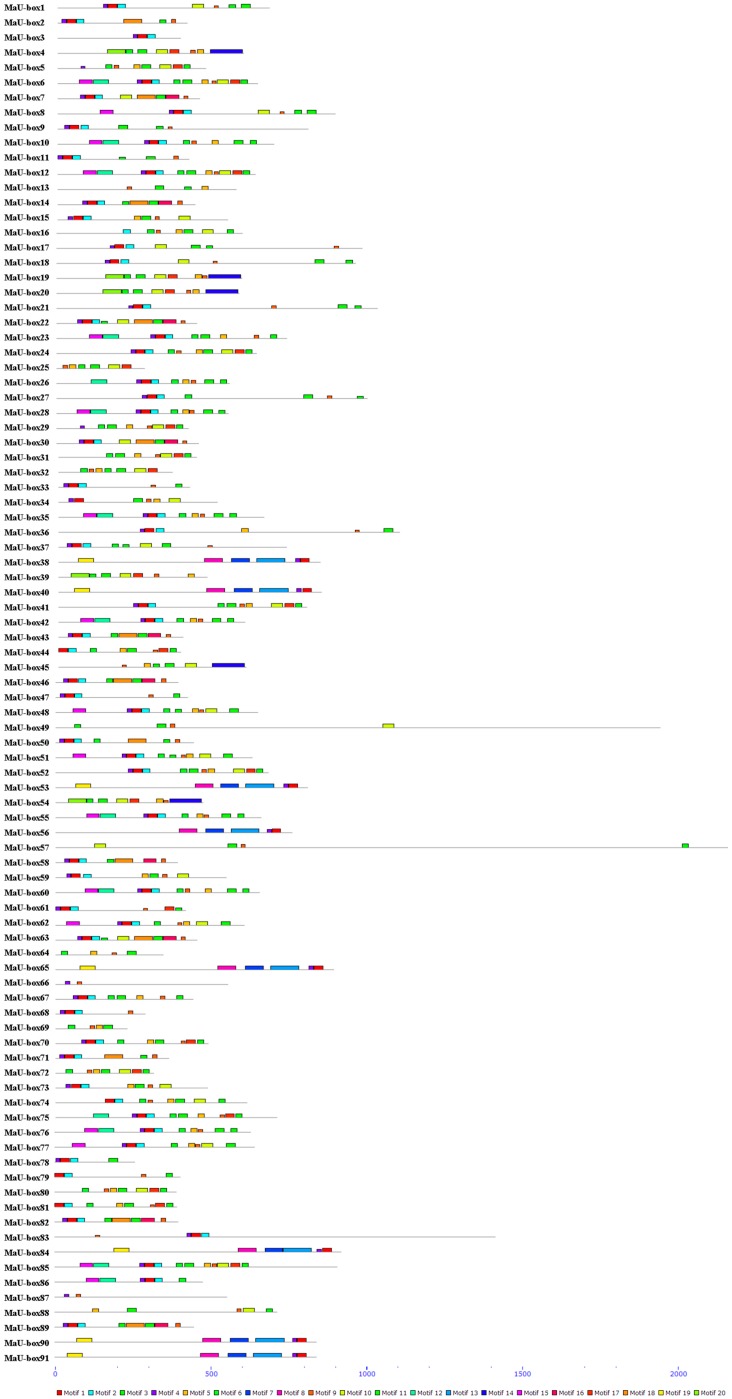
Distribution of conserved motifs for the banana MaU-box proteins.

**Figure 5 ijms-19-03874-f005:**
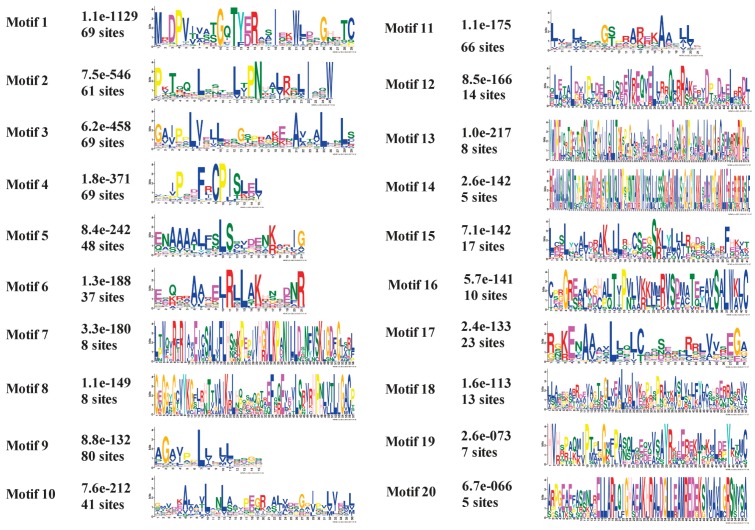
The conserved domains in the MaU-box proteins.

**Figure 6 ijms-19-03874-f006:**
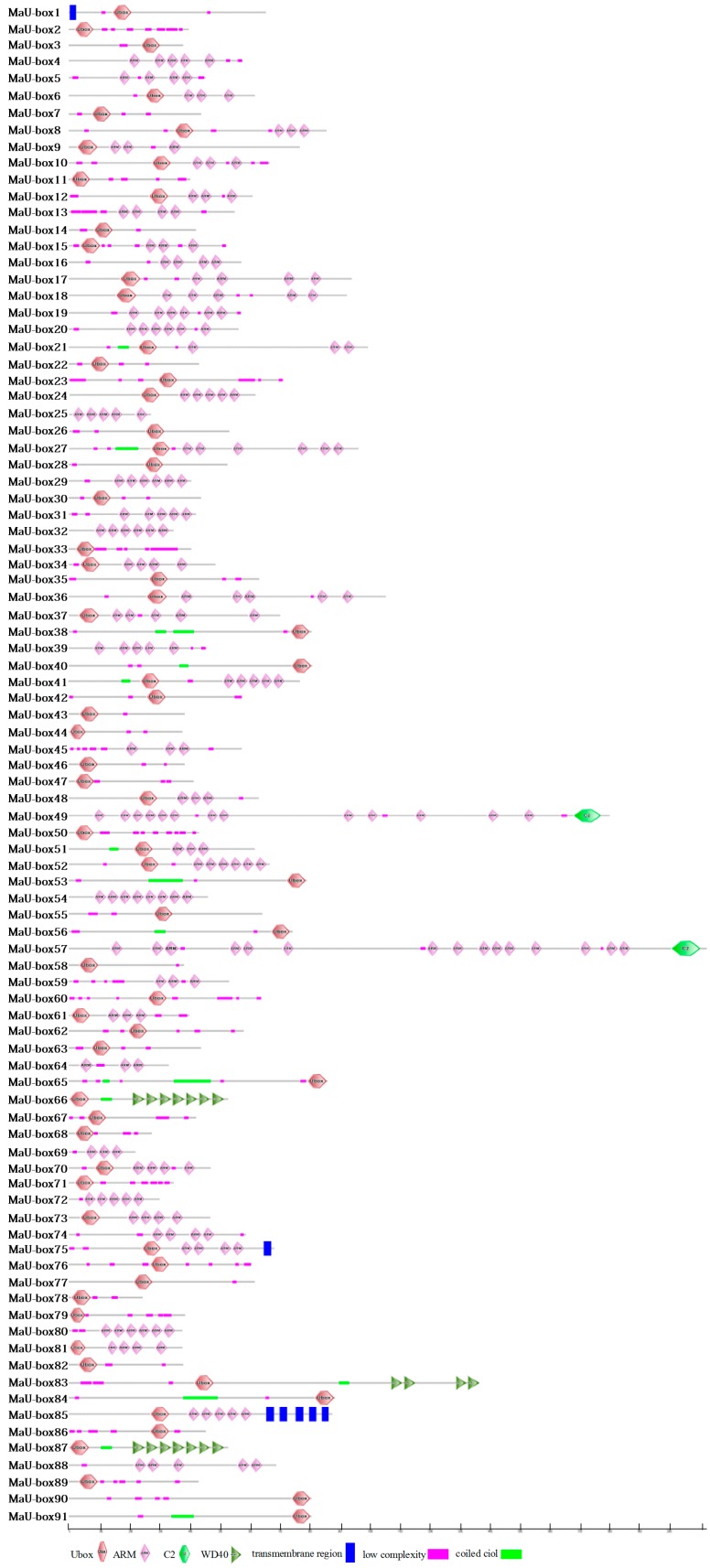
The typical protein structures in the MaU-box proteins.

**Figure 7 ijms-19-03874-f007:**
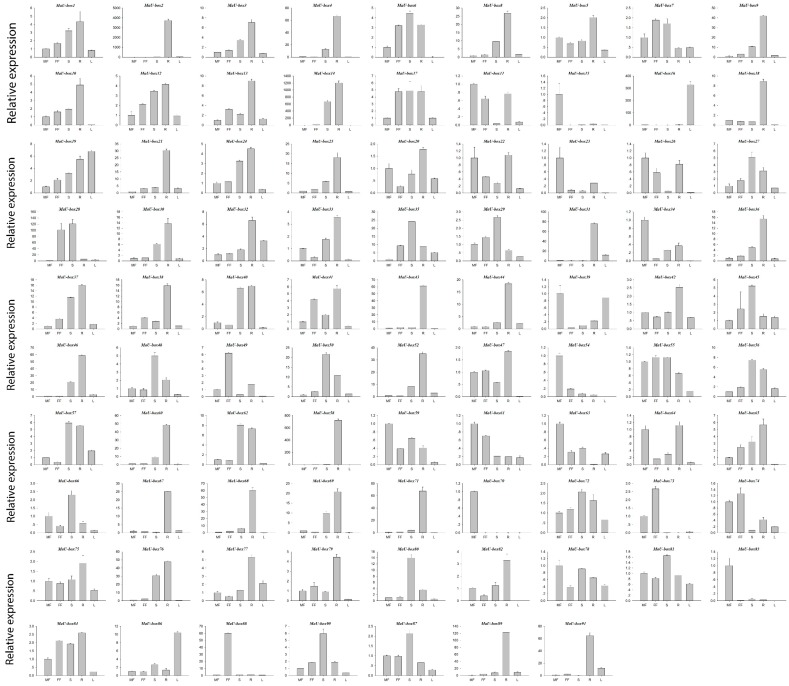
Expression profiles of MaU-box genes in male flowers (MF), female flowers (FF), stems (S), roots (R), and leaves (L).

**Figure 8 ijms-19-03874-f008:**
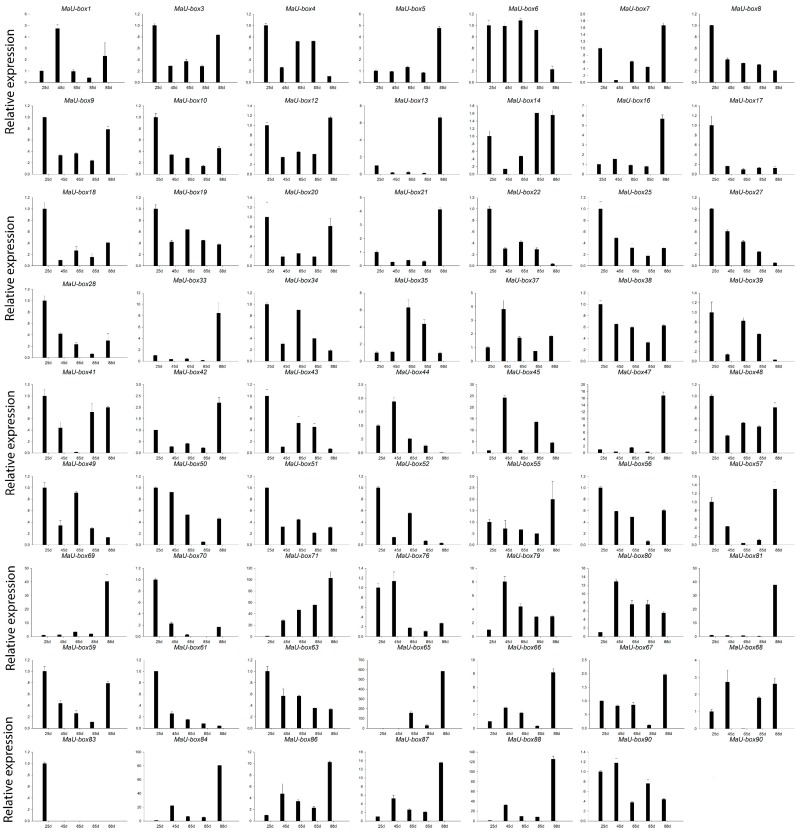
Expression profiles of MaU-box genes in banana pulp during the fruit developmental period. d: day.

**Figure 9 ijms-19-03874-f009:**
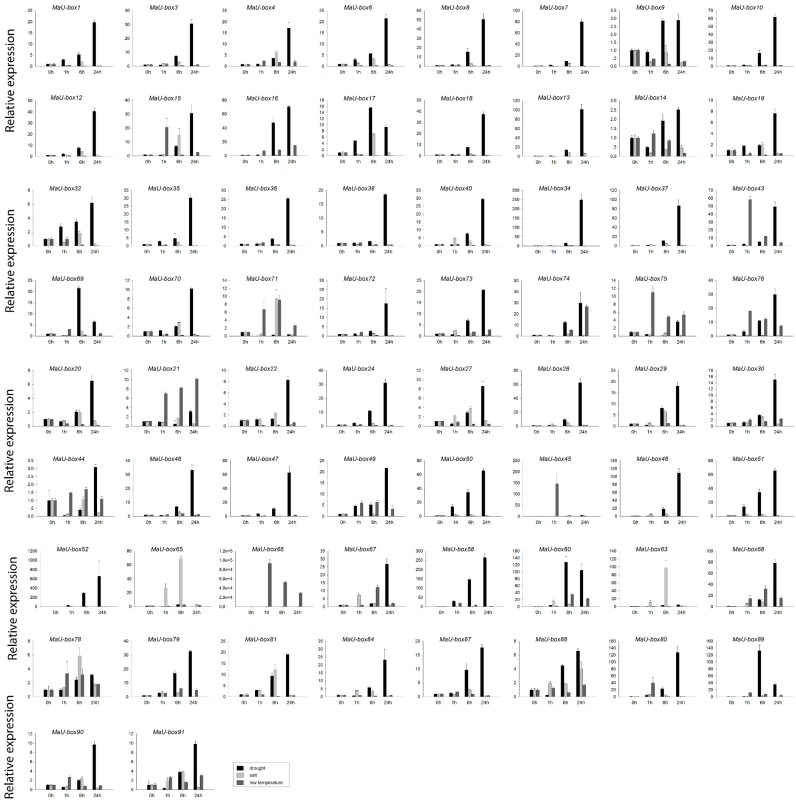
Expression analysis of MaU-box genes in banana leaves under different abiotic treatments.

**Table 1 ijms-19-03874-t001:** The information of Plant U-box (PUB) gene family in banana.

Gene Name	Locus ID	Chromosome Location	Gene DNA (bp)	CDS (bp)	Protein Length (aa)	Molecular Weight (kD)	Theoretical pI	Exon	Putative Localization
MaPUB1	GSMUA_Achr1T12080_001	chr1:9216255..9219037 reverse	2783	1944	647	72.51	5.02	3	Nucleus
MaPUB2	GSMUA_Achr1T13220_001	chr1:10096902..10098167 forward	1266	1191	396	41.13	7.66	2	Nucleus
MaPUB3	GSMUA_Achr1T13730_001	chr1:10477987..10483072 forward	5086	1128	375	42.2	5.58	5	Cytoplasm, Nucleus
MaPUB4	GSMUA_Achr1T16210_001	chr1:12121706..12123412 reverse	1707	1707	568	60.39	8.62	1	Cytoplasm
MaPUB5	GSMUA_Achr1T22790_001	chr1:17158846..17160507 reverse	1662	1362	453	47.14	8.76	3	Nucleus
MaPUB6	GSMUA_Achr1T24290_001	chr1:18636205..18639761 reverse	3557	1836	611	66.29	5.48	6	Nucleus
MaPUB7	GSMUA_Achr2T08620_001	chr2:12553612..12555419 reverse	1808	1305	434	47.08	5.68	2	Nucleus
MaPUB8	GSMUA_Achr2T09640_001	chr2:13094735..13107492 forward	12,758	2547	848	94.33	6.83	6	Nucleus
MaPUB9	GSMUA_Achr2T11370_001	chr2:14269037..14272568 forward	3532	2301	766	84.56	5.85	4	Nucleus
MaPUB10	GSMUA_Achr2T15530_001	chr2:16801361..16803647 reverse	2287	1983	660	72.39	9.22	3	Nucleus
MaPUB11	GSMUA_Achr2T20690_001	chr2:20462606..20463811 forward	1206	1206	401	42.65	6.75	1	Nucleus
MaPUB12	GSMUA_Achr3T00280_001	chr3:253594..258827 forward	5234	1812	603	65.43	5.52	5	Nucleus
MaPUB13	GSMUA_Achr3T01110_001	chr3:844229..845967 forward	1739	1641	546	55.97	9.57	2	Nucleus
MaPUB14	GSMUA_Achr3T02030_001	chr3:1351349..1352952 reverse	1604	1260	419	45.71	8.64	3	Nucleus
MaPUB15	GSMUA_Achr3T06000_001	chr3:4040242..4042320 reverse	2079	1560	519	55.21	7.56	3	Nucleus
MaPUB16	GSMUA_Achr3T06160_001	chr3:4178177..4180306 reverse	2130	1704	567	61.78	9.34	2	Cytoplasm
MaPUB17	GSMUA_Achr3T06320_001	chr3:4302726..4305879 reverse	3154	2805	934	103.26	5.67	5	Nucleus
MaPUB18	GSMUA_Achr3T10610_001	chr3:7831098..7835605 reverse	4508	2745	914	101.97	6.58	5	Nucleus
MaPUB19	GSMUA_Achr3T12370_001	chr3:9151629..9153329 reverse	1701	1701	566	59.93	8.64	1	Cytoplasm
MaPUB20	GSMUA_Achr3T14620_001	chr3:14736420..14738258 reverse	1839	1677	558	59.6	8.58	1	Cytoplasm
MaPUB21	GSMUA_Achr3T19660_001	chr3:20989179..20992456 forward	3278	2946	981	107.65	5.92	4	Nucleus
MaPUB22	GSMUA_Achr3T25750_001	chr3:25831272..25832624 forward	1353	1290	429	46.17	5.38	2	Nucleus
MaPUB23	GSMUA_Achr4T00200_001	chr4:193350..195464 reverse	2115	2115	704	75.88	6.39	1	Nucleus
MaPUB24	GSMUA_Achr4T01530_001	chr4:1265636..1274901 reverse	9266	1836	611	67.06	5.89	6	Cytoplasm, Nucleus
MaPUB25	GSMUA_Achr4T03430_001	chr4:2727458..2728362 reverse	905	810	269	28.36	8.73	2	Nucleus
MaPUB26	GSMUA_Achr4T04900_001	chr4:3844084..3846449 forward	2366	1587	528	59.15	5.65	7	Nucleus
MaPUB27	GSMUA_Achr4T05020_001	chr4:3920776..3926784 forward	6009	2853	950	105.01	5.91	5	Cytoplasm, Nucleus
MaPUB28	GSMUA_Achr4T05030_001	chr4:3927558..3929980 reverse	2423	1575	524	58.48	5.86	7	Nucleus
MaPUB29	GSMUA_Achr4T06990_001	chr4:5215355..5216775 forward	1421	1212	403	41.42	8.56	3	Nucleus
MaPUB30	GSMUA_Achr4T07400_001	chr4:5497755..5499122 reverse	1368	1305	434	47.46	5.63	2	Nucleus
MaPUB31	GSMUA_Achr4T11070_001	chr4:7963314..7965615 reverse	2302	1269	422	43.29	5.56	2	Nucleus
MaPUB32	GSMUA_Achr4T14130_001	chr4:10562237..10563578 forward	1342	1044	347	37.03	5.72	2	Nucleus
MaPUB33	GSMUA_Achr5T01020_001	chr5:597533..598908 forward	1376	1206	401	42.81	8.42	2	Nucleus
MaPUB34	GSMUA_Achr5T05220_001	chr5:3854463..3856370 forward	1908	1458	485	51.15	6.23	3	Nucleus
MaPUB35	GSMUA_Achr5T07140_001	chr5:5158762..5162462 forward	3701	1887	628	69.15	5.75	7	Nucleus
MaPUB36	GSMUA_Achr5T08500_001	chr5:6179256..6185540 reverse	6285	3129	1042	114.11	5.84	5	Nucleus
MaPUB37	GSMUA_Achr5T11040_001	chr5:7966157..7968849 reverse	2693	2094	697	76.13	5.74	4	Nucleus
MaPUB38	GSMUA_Achr5T12380_001	chr5:8892024..8899046 forward	7023	2403	800	90.49	6.39	12	Nucleus
MaPUB39	GSMUA_Achr5T13700_001	chr5:9859570..9861021 forward	1452	1365	454	48.23	8.75	1	Cytoplasm
MaPUB40	GSMUA_Achr5T21060_001	chr5:22947743..22958748 reverse	11,006	2412	803	90.01	5.48	9	Nucleus
MaPUB41	GSMUA_Achr5T21360_001	chr5:23436476..23440174 forward	3699	2280	759	83.62	6.06	4	Cytoplasm, Nucleus
MaPUB42	GSMUA_Achr5T28670_001	chr5:28750592..28755063 reverse	4472	1713	570	62.66	5.75	6	Nucleus
MaPUB43	GSMUA_Achr6T02470_001	chr6:1615103..1616422 reverse	1320	1143	380	40.98	8.69	3	Nucleus
MaPUB44	GSMUA_Achr6T06560_001	chr6:4418997..4420524 forward	1528	1122	373	40.57	7.67	2	Nucleus
MaPUB45	GSMUA_Achr6T10990_001	chr6:7323129..7324847 reverse	1719	1719	572	59.6	8.22	1	Cytoplasm, Nucleus
MaPUB46	GSMUA_Achr6T11310_001	chr6:7517739..7519221 reverse	1483	1143	380	41.25	6.73	3	Nucleus
MaPUB47	GSMUA_Achr6T18140_001	chr6:12201719..12203017 reverse	1299	1230	409	43.26	5.84	2	Nucleus
MaPUB48	GSMUA_Achr6T22980_001	chr6:22176387..22181121 reverse	4735	1884	627	69.88	8.57	10	Nucleus
MaPUB49	GSMUA_Achr6T25390_001	chr6:26469894..26479975 reverse	10,082	5637	1878	202.19	5.28	7	Cytoplasm, Nucleus
MaPUB50	GSMUA_Achr6T25670_001	chr6:26728664..26729950 forward	1287	1287	428	44.86	8.66	1	Nucleus
MaPUB51	GSMUA_Achr6T28500_001	chr6:28868630..28876365 forward	7736	1833	610	67.96	8.79	9	Nucleus
MaPUB52	GSMUA_Achr6T36530_001	chr6:34466958..34471270 reverse	4313	1983	660	72.48	6.42	6	Nucleus
MaPUB53	GSMUA_Achr7T04450_001	chr7:3366660..3370421 forward	3762	2349	798	88.28	7.88	11	Nucleus
MaPUB54	GSMUA_Achr7T05890_001	chr7:4367855..4369228 forward	1374	1374	457	48.06	9.35	1	Cytoplasm
MaPUB55	GSMUA_Achr7T09440_001	chr7:7719776..7722069 forward	2294	1917	638	70.34	8.95	4	Nucleus
MaPUB56	GSMUA_Achr7T09650_001	chr7:7847312..7851814 reverse	4503	2208	735	82.73	8.7	10	Cell membrane, Chloroplast, Cytoplasm, Nucleus
MaPUB57	GSMUA_Achr7T17740_001	chr7:20510968..20523478 reverse	12,511	6279	2092	223.93	5.63	9	Cytoplasm, Nucleus
MaPUB58	GSMUA_Achr7T18640_001	chr7:21413794..21415228 forward	1435	1140	379	41.34	6.41	3	Cell membrane, Nucleus
MaPUB59	GSMUA_Achr7T19940_001	chr7:22941467..22943503 reverse	2037	1590	529	55.7	8.55	3	Nucleus
MaPUB60	GSMUA_Achr7T22130_001	chr7:24687902..24690206 reverse	2305	1902	633	68.33	6.17	3	Nucleus
MaPUB61	GSMUA_Achr7T23310_001	chr7:25579820..25581016 forward	1197	1197	398	43.4	8.95	1	Nucleus
MaPUB62	GSMUA_Achr8T01360_001	chr8:1146290..1151389 forward	5100	1731	576	63.53	8.57	8	Nucleus
MaPUB63	GSMUA_Achr8T04940_001	chr8:3264288..3266015 forward	1728	1299	432	46.78	5.51	2	Nucleus
MaPUB64	GSMUA_Achr8T09590_001	chr8:6746141..6747127 forward	987	987	328	33.95	5.91	1	Nucleus
MaPUB65	GSMUA_Achr8T11510_001	chr8:8320562..8327700 forward	7139	2552	850	95.41	6.12	11	Nucleus
MaPUB66	GSMUA_Achr8T12630_001	chr8:9406796..9419285 forward	12,490	1571	526	57.29	6.33	18	Nucleus
MaPUB67	GSMUA_Achr8T25140_001	chr8:29062381..29066399 reverse	4019	1263	420	44.9	7.71	4	Nucleus
MaPUB68	GSMUA_Achr8T30420_001	chr8:32504280..32505566 forward	1287	822	273	29.04	8.75	3	Nucleus
MaPUB69	GSMUA_Achr9T00690_001	chr9:540580..541705 forward	1126	660	219	23.38	9.33	2	Cytoplasm, Nucleus
MaPUB70	GSMUA_Achr9T01670_001	chr9:1306678..1309170 forward	2493	1401	466	50.56	8.52	2	Nucleus
MaPUB71	GSMUA_Achr9T08750_001	chr9:5638998..5640625 forward	1628	1038	345	36.08	8.47	3	Nucleus
MaPUB72	GSMUA_Achr9T12570_001	chr9:8175348..8180293 reverse	4946	903	300	31.91	6.01	2	Nucleus
MaPUB73	GSMUA_Achr9T14450_001	chr9:9395076..9397076 reverse	2001	1398	465	49.29	8.95	5	Nucleus
MaPUB74	GSMUA_Achr9T14560_001	chr9:9502271..9504578 reverse	2308	1758	585	63.11	8.67	3	Nucleus
MaPUB75	GSMUA_Achr9T15600_001	chr9:10315534..10319034 forward	3501	2031	676	75.01	6.85	7	Nucleus
MaPUB76	GSMUA_Achr9T28820_001	chr9:32677125..32679340 reverse	2216	1800	599	64.89	9.09	4	Nucleus
MaPUB77	GSMUA_Achr9T29420_001	chr9:33126026..33130242 reverse	4217	1836	611	68.18	6.64	9	Nucleus
MaPUB78	GSMUA_Achr10T04850_001	chr10:14480133..14481503 forward	1371	735	244	26.1	4.96	2	Nucleus
MaPUB79	GSMUA_Achr10T09450_001	chr10:19291616..19292842 forward	1227	1155	384	40.37	8.41	2	Nucleus
MaPUB80	GSMUA_Achr10T11310_001	chr10:20821803..20823180 forward	1378	1119	372	39.55	6.28	3	Nucleus
MaPUB81	GSMUA_Achr10T17120_001	chr10:24446918..24449504 reverse	2587	1125	374	40.87	6.14	2	Nucleus
MaPUB82	GSMUA_Achr10T17900_001	chr10:24916493..24917809 forward	1317	1131	376	40.82	8.25	3	Nucleus
MaPUB83	GSMUA_Achr10T22180_001	chr10:27439361..27448280 reverse	8920	4041	1346	148.98	5.49	14	Nucleus
MaPUB84	GSMUA_Achr10T22750_001	chr10:27801994..27810019 forward	8026	2631	876	98.61	6.12	11	Cell membrane, Nucleus
MaPUB85	GSMUA_Achr10T22780_001	chr10:27821009..27830243 forward	9235	2595	864	96.02	8.5	14	Nucleus
MaPUB86	GSMUA_Achr10T27710_001	chr10:30760030..30761828 reverse	1799	1359	452	50.07	5.87	5	Nucleus
MaPUB87	GSMUA_Achr11T02530_001	chr11:1736304..1752382 reverse	16,079	1581	526	57.22	6.41	18	Nucleus
MaPUB88	GSMUA_Achr11T06020_001	chr11:4516270..4520590 reverse	4321	2040	679	73.94	7.63	3	Nucleus
MaPUB89	GSMUA_Achr11T09460_001	chr11:7338409..7341969 forward	3561	1278	425	45.72	8.17	3	Nucleus
MaPUB90	GSMUA_Achr11T20980_001	chr11:21780963..21789676 reverse	8714	2400	799	89.46	5.71	9	Nucleus
MaPUB91	GSMUA_AchrUn_randomT07480_001	chrUn_random:32367065..32372845 forward	5781	2403	800	90.95	6.37	11	Nucleus

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
