# Peer review of "Genome-Wide Identification and Analysis of U-Box E3 Ubiquitin-Protein Ligase Gene Family in Banana"

_ijms, 2018, doi:10.3390/ijms19123874_

Round 1

Reviewer 1 Report

After the authors revised the manuscript, it still raises numerous concerns, some of them were mentioned in my first review and were not addressed:

1. In the abstract it is written that U-box gene family is known as ubiquitine ligase E3, but U-box proteins are actually a type of E3-Ubiquitin ligases. In line 10, it is not clear what does mean by accumulating abnormal proteins. In line 22 I do not understand what do the authors mean by understanding the construction.

2. The authors still claim that the genes are located in the chromosomes, which is normal. But now they claim that 90 out of 91, where is located the one that is NOT located in a chromosome?

3. As mentioned in my fisrt review, in the introduction the authors do not mention a very recent and comprehensive review about U-box E3 ligases in plants (doi: 10.1093/jxb/erx411). Similarly, the introduction is lacking reference to the important role of these proteins in Arabidopsis.

4. In line 100, there are 22 genes that contain an ARM conserved motif. Do these genes do not contain and U-box motif? These need to be clarified.

5. In line 113 it would be more accurate referring to all tissues tested and specifying which tissues where tested.

6. The discussion has to be rewritten. As now is just a list of results from previous work without connexion and sense.

7. The methods for the complete expression experiments are missing.

8. While in the results is mentioned all the tissues in the material and method the authors talk about various tissues. The tissues tested should be clarified.

9. The figures are still missing explicative figure legends, for example in expression figures, which are the tissues in which the expression is shown.

Author Response

1. In the abstract it is written that U-box gene family is known as ubiquitine ligase E3, but U-box proteins are actually a type of E3-Ubiquitin ligases. In line 10, it is not clear what does mean by accumulating abnormal proteins. In line 22 I do not understand what do the authors mean by understanding the construction.

Response: Thank you for the useful comments and suggestions on our manuscript. ‘The U-box gene family, also known as ubiquitin ligase E3, is a family of genes with a U-box domain’  has been changed to ‘The U-box gene family is a family of genes which encode U-box domain-containing proteins ’in P.5, L. 92-93. We deleted the unclear description ‘U-box genes respond to cellular stress by accumulating abnormal proteins’ and ‘construction’.

2. The authors still claim that the genes are located in the chromosomes, which is normal. But now they claim that 90 out of 91, where is located the one that is NOT located in a chromosome?

Response: Thank you for the useful comments. Some of the sequences in genome sequencing are unsequenced, so it is not known on the one gene belong to which chromosome.

3. As mentioned in my fisrt review, in the introduction the authors do not mention a very recent and comprehensive review about U-box E3 ligases in plants (doi: 10.1093/jxb/erx411). Similarly, the introduction is lacking reference to the important role of these proteins in Arabidopsis.

Response: Thank you for the useful suggestions on our manuscript. We added the important article (doi: 10.1093/jxb/erx411) about U-box E3 ligases in plants to our manuscript, and the paper revealed the important role of U-box E3 ligases (e.g.,Inactivation of the Arabidopsis PUB13 also results in spontaneous cell death, enhanced levels of the defence hormone SA, and early flowering).

4. In line 100, there are 22 genes that contain an ARM conserved motif. Do these genes do not contain and U-box motif? These need to be clarified.

Response: Thank you for the useful suggestions. ‘Among the 91 U-box genes, 45 (50%) contained U-box conservative motifs, 22 (24%) contained ARM conservative motifs, and 24 (26%) contained both U-box conserved motifs and ARM conserved motifs’ has been changed to ‘Among the 91 U-box genes, 45 (50%) contained U-box conservative motifs without ARM motifs, 22 (24%) contained ARM conservative motifs without U-box motifs, while 24 (26%) contained both U-box conserved motifs and ARM conserved motifs.s’.

5. In line 113 it would be more accurate referring to all tissues tested and specifying which tissues where tested.

Response: Thank you for the useful suggestions. We added the tissue name (young roots, young stems, young leaves, female flowers and male flowers) in result description and figure caption.

6. The discussion has to be rewritten. As now is just a list of results from previous work without connexion and sense.

Response: Thank you for the useful suggestions. We have rewritten the discussion.

7. The methods for the complete expression experiments are missing.

Response: We have added the methods for the complete expression experiments: ‘The MaActin fragment of the banana was selected as the internal reference, and the primers were designed according to the registered sequence. All the MaU-box genes secific primers were designed according to the coding sequences by Primer5 software and checked using Blast. The relative expression level of the U-box gene was calculated using Equation 2-ΔΔCT.

8. While in the results is mentioned all the tissues in the material and method the authors talk about various tissues. The tissues tested should be clarified.

Response: Thank you for the useful suggestions. We change the sentence ‘The Mau-box gene family was differentially expressed in various tissues’ to ‘The Mau-box gene family was differentially expressed in male flowers (MF),female flowers (FF), stems (S),roots (R), and leaves (L ) ’.

9. The figures are still missing explicative figure legends, for example in expression figures, which are the tissues in which the expression is shown.

Response: Thank you for the useful suggestions. We have corrected the unclear figure legends. (1)‘Figure 7. Expression profiles of MaU-box genes in various organs’ has been changed to ‘Figure 7. Expression profiles of MaU-box genes in male flowers (MF),female flowers (FF), stems (S),roots (R), and leaves (L )’. (2) ‘Figure 8. Expression profiles of MaU-box genes during the fruit developmental period’ has been changed to ‘Figure 8. Expression profiles of MaU-box genes in banana pulp during the fruit developmental period’. (3) ‘Figure 9. Expression analysis of MaU-box genes under different abiotic treatments.’ has been changed to ‘Figure 9. Expression analysis of MaU-box genes in banana leaves under different abiotic treatments.’

Reviewer 2 Report

The paper describes an important topic, regarding the identification and characterization of E3 Ubiquitination ligase genes. While the MS is well written and the results clearly presented, it is my opinion that the results are only superficially discussed, and that the discussion section should be extensively revised and extended. Some highlights:

The authors performed the phylogenetic analysis of the 91 U-box proteins and included rice and Arabidopsis homologues. In the discussion soybean genes are also referred. However, the implications of the variation in the number of genes as well as the diversity of U-box proteins within a species and between species should be also discussed (and for that more comparisons can be included); how does it relates to the interaction of E3 ubiquitin ligases with their substrates for ubiquitination? what is the significance of the finding that a complete U-box 158 domain was found in all PUB proteins?

Regarding the exon/intron number: what is the relevance from the evolutionary point of view? is this number higher or lower than that of related and unrelated species?

Can you extend the discussion on the hypothesis about mutations during the evolution of U-box genes from banana? also please point other examples

Differential gene expression: please extend the discussion regarding the putative role of each protein/group of proteins in each biological process and perform more dynamic comparison with what has been described in other species.

The Materials and methods section lacks the methodology used to analyse gene expression.

Author Response

1. The paper describes an important topic, regarding the identification and characterization of E3 Ubiquitination ligase genes. While the MS is well written and the results clearly presented, it is my opinion that the results are only superficially discussed, and that the discussion section should be extensively revised and extended.

Response: Thank you for the useful suggestions. We have written the discussion by the suggestions.

2. The authors performed the phylogenetic analysis of the 91 U-box proteins and included rice and Arabidopsis homologues. In the discussion soybean genes are also referred. However, the implications of the variation in the number of genes as well as the diversity of U-box proteins within a species and between species should be also discussed (and for that more comparisons can be included); how does it relates to the interaction of E3 ubiquitin ligases with their substrates for ubiquitination? what is the significance of the finding that a complete U-box 158 domain was found in all PUB proteins?

Response: Thank you for the useful suggestions. We discussed the diversity of different MaU-box genes: ‘In this study, most banana U-box proteins show closer phylogenetic distance to their putative banana homologs than to their corresponding putative rice and Arabidopsis orthologs. Moreover the U-box from banana had a closer relationship with rice compared with Arabidopsis. Interestingly, banana proteins MaU-box56, MaU-box78, MaU-box83 and MaU-box84 showed closer phylogenetic relationship to the rice proteins than to their banana paralogs, suggesting that these banana proteins and their corresponding rice orthologs have evolved from a common ancestor before the speciation of the two species’. And we discussed the effect of gene different structure on the interaction of E3 ubiquitin ligases with their substrates for ubiquitination: ‘The ARM repeats have been shown mostly to mediate the interaction with substrates, indicating that interaction renders substrates available for ubiquitination. So the U-box proteins without ARM repeats in banana might have different interaction of E3 ubiquitin ligases with their substrates compared with the U-box proteins containing ARM repeats.’ We think the significance of the finding as follow: ‘different protein stucture is relates to the interaction of E3 ubiquitin ligases.’

3. Regarding the exon/intron number: what is the relevance from the evolutionary point of view? is this number higher or lower than that of related and unrelated species?

Response:Thank you for the useful suggestions. We read the references on exon/intron carefully, and found the number of exon/intron is not related to evolutionary gene. So we deleted the unclear point on exon/intron number.

4. Can you extend the discussion on the hypothesis about mutations during the evolution of U-box genes from banana? also please point other examples.

Response: Thank you for the useful suggestions. We extend the disscussion on the hypothesis about mutations: ‘Interestingly, banana proteins MaU-box56, MaU-box78, MaU-box83 and MaU-box84 showed closer phylogenetic relationship to the rice proteins than to their banana paralogs, suggesting that these banana proteins and their corresponding rice orthologs have evolved from a common ancestor before the speciation of the two species’ . And we pointed the rice examples.

5. Differential gene expression: please extend the discussion regarding the putative role of each protein/group of proteins in each biological process and perform more dynamic comparison with what has been described in other species.

Response:Thank you for the useful suggestions. We added the examples on putative role of each protein/group of proteins and  perform more dynamic comparison: ‘In the present study, the  29 MaU-box genes (MaU-box3/4/8/9/10/13/17/18/19/20/22/25/27/28/34/38/39/41/43/48/49/50/51/52/56/59/61/63/70) had the highest expression at the beginning of banana’s developmental stage (day 25), which could be explained by the fact that the highly expressed genes usually play important roles in plant development’. And ‘OsPUB57 showed a stronger expression only in the resistant plants carrying the Pi9-resistant gene [17]. Consistent with this study, in our study, the MaPUB84 and MaPUB91 genes which showed closer phylogenetic distance to OsPUB57 had high expressions under stress . ’

6.The Materials and methods section lacks the methodology used to analyse gene expression.

Response: Thank you for the useful suggestions. We added the methodology used to analyse gene expression: ‘The MaActin fragment of the banana was selected as the internal reference, and the primers were designed according to the registered sequence. All the MaU-box genes secific primers were designed according to the coding sequences by Primer5 software and checked using Blast. The relative expression level of the U-box gene was calculated using Equation 2-ΔΔCT.

Round 2

Reviewer 1 Report

I would like to thank the authors for taking in consideration my comments.

Author Response

English language and style are fine/minor spell check required.

Response: We have improved English language to make it more clear and read more smoothly. Thank you very much!

Reviewer 2 Report

After reading the revised version of your MS, it is my opinion that the paper is now acceptable for publication. However, I would suggest minor language and text editing.

Author Response

Response: We have improved English language to make it more clear and read more smoothly. Thank you very much!